# E-Cardiac Care: A Comprehensive Systematic Literature Review

**DOI:** 10.3390/s22208073

**Published:** 2022-10-21

**Authors:** Umara Umar, Sanam Nayab, Rabia Irfan, Muazzam A. Khan, Amna Umer

**Affiliations:** 1School of Electrical Engineering and Computer Science (SEECS), National University of Sciences and Technology (NUST), Islamabad 44800, Pakistan; 2Department of Computer Sciences, Quaid i Azam University, Islamabad 45320, Pakistan; 3Department of Computational Sciences, The University of Faisalabad (TUF), Faisalabad 38000, Pakistan

**Keywords:** arrhythmia, artificial intelligence (AI), cardiac, communication technologies, Electrocardiogram (ECG), systematic literature review (SLR)

## Abstract

The Internet of Things (IoT) is a complete ecosystem encompassing various communication technologies, sensors, hardware, and software. IoT cutting-edge technologies and Artificial Intelligence (AI) have enhanced the traditional healthcare system considerably. The conventional healthcare system faces many challenges, including avoidable long wait times, high costs, a conventional method of payment, unnecessary long travel to medical centers, and mandatory periodic doctor visits. A Smart healthcare system, Internet of Things (IoT), and AI are arguably the best-suited tailor-made solutions for all the flaws related to traditional healthcare systems. The primary goal of this study is to determine the impact of IoT, AI, various communication technologies, sensor networks, and disease detection/diagnosis in Cardiac healthcare through a systematic analysis of scholarly articles. Hence, a total of 104 fundamental studies are analyzed for the research questions purposefully defined for this systematic study. The review results show that deep learning emerges as a promising technology along with the combination of IoT in the domain of E-Cardiac care with enhanced accuracy and real-time clinical monitoring. This study also pins down the key benefits and significant challenges for E-Cardiology in the domains of IoT and AI. It further identifies the gaps and future research directions related to E-Cardiology, monitoring various Cardiac parameters, and diagnosis patterns.

## 1. Introduction

Following the available information as confirmed by the World Health Organization (WHO) [1], cardiovascular disease claims a large number of causalities across the globe [2] and is responsible for approximately 80% of sudden deaths. Moreover, in more than 15% of the deaths, cardiac arrhythmia is considered the chief reason. Thus, promoting cardiovascular health is vital and requires an overhaul of healthcare systems [3].

The rapidly expanding Internet of Things (IoT) [4] technology has the capability to monitor and control the critical human functions, irrespective of where the individual is located or what they are doing. Medical IoT (MIoT) is a cutting-edge technology that functions by exploiting the advantages of the Internet at a very affordable cost with minimum effort. The MIoT-based cardiac system guarantees monitoring the physical symptoms [5] of cardiac patients, such as temperature, Blood Pressure (BP), Oxygen Saturation (SPO2), Electrocardiogram (ECG), Heart Rate (HR) [6], and linked environmental parameters effectively and without any failure. The MIoT cardiac care framework is a customized paradigm that meets the requisite medical and safety standards of pervasive cardiac healthcare, including serious heart-related issues.

Various cardiac (heart) abnormalities can be detected through an Electrocardiogram (ECG) which is a medical testing platform that keeps track of electrical activity the heart generates as it contracts. An electrocardiograph is a device that records patient’s ECG. An ECG is a valuable tool for identifying problems associated with heart rate or heart rhythm. It offers assistance to the physician in determining whether a patient is having a heart attack or has had one in the past. An ECG is usually the first option for a cardiac test because of its proven dependability. An ECG is helpful to determine if one’s pulse is difficult to feel (bradycardia), or it is too fast (tachycardia) to count accurately. An ECG can also show heart rhythm irregularities, i.e., arrhythmia. The main types of arrhythmia are mentioned in Table 1.

Similarly, atrial fibrillation, atrial flutter, and premature or extra beats are the other types of cardiac issues. Figure 1 shows waveforms for different arrhythmia types. Atrial fibrillation refers to a rapid, disorganized, and irregular heart rhythm., while atrial flutter is an atrial arrhythmia generated by a fast circuit in the atrium. Compared to atrial fibrillation, atrial flutter is typically more organized and regular.

A comprehensive review of E-Cardiology, which encompasses the Internet of Things, artificial intelligence, and cardiology could help understand the essential building blocks of an IoT-based cardiac care system and intelligent diagnosis of various cardiovascular diseases. It can also help to develop a complete picture of various hardware devices (sensors), AI techniques, and communication technologies adopted by the existing studies in the field of intelligent cardiac healthcare.

Following the introduction, Section 2 of this paper briefly discusses related works; Section 3 highlights the contributions made by this paper. Section 4 elaborates on the review methodology adopted for conducting the survey. Section 5 gives the outcomes of the selected studies with a detailed analysis of the research questions (RQs). This section is further divided into four subsections. Section 6 contains a discussion. Section 7 summarises the conclusions.

## 2. Related Works

This section presents a brief explanation of the related surveys in the field of IoT-based cardiac healthcare.

The primary goal of the study [7] was to collect the latest facts, figures, and evidence on the use of preprocessing techniques for heart disease classification. The review study also summarised the impact of the most frequently used preprocessing tasks and techniques and the performance in the field of cardiology. This review paper covered the literature from 2000 to June 2019.

A survey on IoT and AI in healthcare was presented by [8] for 2007 to February 2018. The paper highlighted the top application classifications, which included wearables, sensor networks, connectivity options, and disease detection and treatment. This review identified gaps and provided future research directions related to technology and design. However, this survey analysed only three online databases.

A review article on data mining techniques frequently used in the field of cardiology until 2015 was presented in [9]. The performance comparison of various data mining models in cardiology were also discussed in this review paper.

The authors in [10] presented a survey on the Internet of Things (IoT) for healthcare using mobile computing. This systematic study investigated how mobile computing assisted IoT in a healthcare environment. Moreover, the intention of this paper was to analyse the impact of mobile computing on IoT technology in Smart hospitals and the field of healthcare. This study covered the literature between 2011 and 2019.

Another study [11] proposed a substantial review of various IoT applications in a life-saving environment, as well as various other fields in Smart cities. It also contrasted IoT with M2M and highlighted some drawbacks of IoT technology. This review article covered 2013 to 2018 through the Scopus database.

Another study [12] presented literature on (IoT) technologies and several projects for healthcare in 2018. This paper provided a review of primary medical IoT sensors and an overview of state-of-the-art IoT infrastructure essential for healthcare. It focused on the latest IoT technologies for healthcare services, such as cloud computing, big data, RFID, WSN, Bluetooth, Wi-Fi, and other vital medical sensors. However, this study lacks a systematic review.

The study [13] highlighted various IoT applications and was presented in 2022. The study focused on IoT adoption in Pakistan and France in 2020. This systematic study highlighted the barriers and possibilities for the implementation of IoT applications. It also indicated the influence of COVID-19 on IoT adoption in the healthcare domain.

The [14] systematic review discussed telemedicine and healthcare IoT (HIoT). It covered 146 articles between 2015 and 2020. The articles were divided into five categories after a technical analysis. In addition to the benefits and limitations of the selected methods, a comprehensive comparison of evaluation techniques, tools, and metrics was also included. This study presented a summary of healthcare applications of IoT (HIoT).

The discussion so far is limited to only a particular aspect of Smart healthcare/E-Cardiology and does not genuinely attempt to cover the domain holistically. When we say “entire domain”, it means AI-based IoT, which encompasses preprocessing techniques and also various communication technologies. According to the deficiencies of the existing review papers, we provide a comprehensive systematic literature review for the following reasons:The latest research articles need to be covered to assess the current state of the art.The present studies do not cover all the aspects of E-Cardiology.

The following section highlights the contributions made by this review study, thus bringing novelty to this systematic review study.

## 3. Contributions

This review paper highlights the influence of IoT, communication technologies, AI models, and preprocessing techniques in cardiac healthcare using our review protocol. Moreover, this study covers the complete and latest infrastructure for E-Cardiology, including its benefits and challenges. Thus, this systematic review covers almost all aspects of E-Cardiology which have not been discussed before in such a comprehensive way under one umbrella.The study presents the systematic analysis of the most recent studies (2016 to 2021) to investigate our formulated research questions.This paper incorporates monitoring of vital CCU parameters, ECG analysis, and classification of various heart disorders, thus giving a thorough picture of E-Cardiology.This review study provides recommendations and future guidelines for researchers and cardiologists as well.

The next section discusses the research methodology adopted for our SLR.

## 4. Review Methodology

A systematic literature review (SLR) paradigm is followed in this paper for reviewing papers from the most reliable resources, as shown in Figure 2. Principally, the research work, applications, and monitoring/detection techniques provided by AI-aided MIoT in cardiac care are considered. The primary studies have been then passed through a quality assessment process for the study analysis to produce the best fit results.

The following subsections briefly describe the detail of each step involved in our review protocol.

### 4.1. Defining Review Strategy

The application of medical IoT in cardiac care is a compelling field of study for the researchers, so the primary focus of this SLR was to formulate the research questions exploring how medical IoT is affecting cardiac care and the significance of artificial intelligence in the diagnosis and detection of various heart diseases.

The review questions in Table 2 indicate how MIoT and AI are contributing to cardiac healthcare systems in Smart hospitals.

### 4.2. Defining Search Strategy

Once the research questions were designed, the next step was to indicate and state the search strategy to be followed precisely. Therefore, the primary literature mentioned in Appendix A (Table A1) was identified using three search strings which were used in the five digital databases, namely IEEE Xplore, ACM Digital Library, SpringerLink, ScienceDirect, and Google Scholar. These are the most popular online data resources in the domain of computer science and information technology. Second, these digital libraries were used as sources for previous systematic literature reviews related to computer science and E-Cardiology [15].

Our search span included the period of 2016 to 2021. The criteria used for the selection of search terms or keywords is mentioned below [16]:The important terms were extracted from the research questions.Synonyms and alternate spellings were identified for the key terms.Keywords were identified from various books and relevant research articles.For synonyms or alternating spellings, the Boolean operator OR was used.Boolean AND operator was used to interlink significant terms.

After the critical analysis of the key terms, three search strings were formed in order to extract the relevant information. These search strings were checked on each of the aforementioned databases by changing their patterns to retrieve the best relevant results. The three search strings are given in Table 3.

### 4.3. Inclusion and Exclusion Criteria

To identify and include studies relevant to answer the RQs, inclusion and exclusion criteria were developed as described in the section “Defining Review Strategy”. To find the most appropriate publications, we defined the inclusion and exclusion criteria as mentioned in Table 4.

The authors evaluated each forthcoming paper to decide whether it should be included or excluded. The selection of papers was accomplished by following the three steps mentioned below.

The first step included the removal of duplicated and redundant papers; perusing the keywords, abstracts, and titles of research articles was the next step. Reading of full length research papers was carried out in the last step. Accordingly, the inclusion and exclusion criteria were implemented to their full effect. The articles that attracted difference of opinion were discussed and reviewed again by the authors, either using the full text or the partial text, until a consensus was achieved on an agreed-upon draft.

### 4.4. Quality Assessment Criteria

In this step, based upon the coherence and relevancy, we analyzed all the collected studies to address the defined research questions. A deep analysis of each paper was made, and based on our research questions, 134 papers were selected. Out of those 134 research papers, the papers having considerable citation count, appearing in good impact factor journals, and being delivered at highly ranked conferences were finally selected, thus leaving a total of 104 papers for the review, shown in Appendix A (Table A1).

### 4.5. Quantitative Analysis

The last step of our review protocol design was conducted to execute necessary statistical analysis on quantitative data. In this step, we quantitatively summarised and analyzed the results extracted from various sources such as conferences, journals, and book sections. Then, we carried out some quantitative statistical analysis of the findings to explore more about our research questions (RQs) and trends.

Figure 3 gives a thorough overview of our screening and assessment method for the statistical analysis of our literature. Five databases were chosen for the review, as illustrated in this figure. A total of 502 documents were chosen for review and analysis. The majority of papers were discovered to be duplicates. Thus, 203 records were eliminated before screening. Papers were removed for a few different reasons. Articles were chosen in the screening process based on a planned inclusion and exclusion strategy. Following the screening, 104 papers were chosen based on inclusion and exclusion standards.

The next section highlights the “Outcomes” of this systematic review.

## 5. Outcomes

### 5.1. RQ 1: What Are the Vital Hardware Components/Sensors Used in E-Cardiac Architecture for Different CCU Parameters?

The cardiac healthcare monitoring system in an IoT sphere encompasses the various IoT sensory modules and technologies attached to the patient, receiving sensory data, and sending data to the cloud for further monitoring, processing, and decision making. In an IoT-based cardiac healthcare monitoring system, the sensors, such as heart rate/pulse sensor, temperature sensor, blood pressure sensor, blood oxygen sensor, and ECG sensor, obtain sensory parametric values from the patient, transmit data through specific communication technologies to the cloud, apply machine learning practices to the learned parametric values, and generate alerts to the specialist suggesting timely action when warranted.

**(i)** 
**Heart Rate Sensory Unit**


Heart rate monitoring plays a crucial role in patient cardiac abnormalities diagnosis, detection, and classification. Several cardiac ailments and disorders occur due to a patient’s high or low heart rate. Normal beats per minute (bpm) are 60–100. Less than 60 bpm is considered to be low and greater than 100 is considered to be high bpm. We discovered comprehensive studies that used several types of heart beat sensors for bpm monitoring. The studies [17,18] used heart beat pulse sensors to measure patient heart rates in a real-time environment. In this article [19], the KY 093 module was used to obtain heart rate values. Using a MAX30100 pulse oximeter, the authors in [20] collected heart beat information. The publications [21,22,23] utlized an ECG module AD8232 to obtain the patients heart rate data in real time for monitoring purposes. Table 5 mentions recent studies on heart rate sensors. Each heart rate sensor has its own set of properties. This table shows some important characteristics of several heart beat sensor variants such as pins, type, operating voltage, low current supply, accuracy, and so on.

**(ii)** 
**Temperature Sensory Unit**


Body temperature is an essential parameter for the development of cardiac healthcare monitoring. Various analog and digital temperature sensors are available for determining body temperature. Temperature can be measured in celsius or fahrenheit. Temperatures above 37.5 or 38.3 celsius are considered high. The temperature sensor LM35 is referenced in [24,25,26,27] for health monitoring. The authors in [18,28] used an 18DS20 sensor for temperature monitoring in a real-time environment. Table 6 shows several of the temperature sensors, along with descriptions. The LM35 and 18DS20 sensors are the most widely employed temperature sensors in the research studies that we analysed.

**(iii)** 
**Blood Pressure (BP) Sensory Unit**


BP monitoring is a fundamental biological measure for the detection and diagnosis of cardiac incongruities and anomalies. BP values can be obtained using various sensory units and devices. Systolic and diastolic values are captured by BP sensors or devices to be examined by a physician. Normal BP is less than 120/80 mmHg, while low BP, called “hypotension”, is below 90/60 mmHg, and high BP, called “hypertension”, is above 140/90 mmHg. Our research discovered a publication on E-Cardiology that dealt with cardiac patients’ BP. In 2017 [29], a digital BP monitor (OMRONHBP1300) was used to monitor and automatically detect cardiac arrhythmia. The paper published in 2019 [30] examined predicting cardiac ailments in E-Cardiology using ECG, cholesterol, and BP. The MPX10 BP sensor was utilised in the 2020 publication [27] for patient health monitoring. Table 7 shows lists of sensors and devices utilized in the past few years for BP monitoring of cardiac patients, along with their comparable attributes.

The multiple modules of BP sensors and devices used in past studies, as well as different sensors of other cardiac parameters, are shown in Table 7.

**Table 5 sensors-22-08073-t005:** Features of heart rate sensors/devices.

Heart Rate/ Pulse Rate Sensors	Features							
	Major Pins	INT	Type	Operating Voltage	BPM	Low Supply Current	Electrodes Configuration	Acc
**Pulse Sensor [31]**	GND, Vcc, Signal	N	IR LED (Analog)	3.3 V to 5.0 V	Y	N/A	N	N/A
**AD8232 [32]**	GND, 3.3 V, Output, LO+, LO−, SDN	Y	IR LED (Analog)	3.6 V	Y	170 µA (typical)	Y	N/A
**KY-039 [33]**	GND, Vcc, Signal	N	IR LED (Analog)	5 V	Y	N/A	N	N/A
**Holter Device [34]**	3/5/12 Electrodes	Y	Digital Device	N/A	Y	N/A	Y	N/A
**SpO2 Sensor** **device [35]**	Fingertip Sensor	Y	IR LED (Analog)	N/A	Y	N/A	N	±2% for SPO2, ±2 bpm for Pulse Rate
**MAX30100 Pulse** **Oximeter and** **Heart Sensor [36]**	VIN, SCL, SDA, INTERRUPT, IRD, RD, GND	Y	Int IR LED, Photo Sensor	1.8 V and 3.3 V	Y	170 µA, (typical)	N	98.84% for SPO2, 97.11% for Heart Rate

Acc—accuracy; BPM—beats per minute; GND—ground; INT—integrated IR; LED—infrared light—emitting diode; IRD—IR LED to driver; LO—leads off; N/A—not applicable; N—no; RD—red LED to driver; SCL—serial clock; SDA—serial data; SDN—shutdown control input; V—voltage; VCC—voltage common collector; VIN—voltage input; Y—yes.

**(iv)** 
**Oxygen Sensory Unit**


Blood oxygen can be monitored using several IoT-based blood oxygen sensory units, such as pulse oximetry sensors, to obtain oxygen saturation levels along with a patient’s heart rate. Ready-made wearable devices are also available to measure blood oxygen saturation levels. Blood oxygen is measured in percentage. The normal blood oxygen saturation is 90 to 100%. The study [37] describes a pervasive healthcare monitoring service system that uses an SpO2 device to measure oxygen saturation. The MAX30100 pulse oximeter has proven to be useful in measuring blood oxygen levels in cardiac patients [20]. Our findings and the literature on IoT-based cardiovascular healthcare monitoring used the sensors mentioned in Table 8 to measure oxygen saturation. This table lists several important and common features of oxygen saturation sensors and devices, such as addressed parameters, voltage, type, accuracy, pins, range, and so on. Table 8 shows that blood oxygen sensors/devices are used for cardiac patients in very few studies.

**(v)** 
**ECG Unit**


ECG is the most crucial biological parameter for monitoring, detecting, predicting, and classifying cardiac irregularities and variations in the human heart. The ECG AD8232 module was used in the studies [21,22,38,39,40] to monitor ECG and detect cardiac anomalies in cardiovascular patients. Heart abnormalities were detected with the use of the ECG AD8233 module [41]. In Table 9, recent papers published on multiple ECG sensors and devices are mentioned along with their necessary and comparable features. Low supply current, electrodes, the sampling rate, right leg drive shut down, single supply operation, high pass filter, output, operating temperature, pins, and other features of various ECG modules are considered as the most notable attributes.

Table 10 shows detailed and comprehensive literature analysed to find IoT-based cardiovascular sensors and devices used in previous studies from 2016 to 2021. This table demonstrates that the majority of research on ECG has been conducted using the ECG AD8232 module to detect anomalies in cardiac patients.

**Table 6 sensors-22-08073-t006:** Features of temperature sensors/devices.

Temperature Sensor	Features						
	Type	${}^{{}^{\circ}}$C/${}^{{}^{\circ}}$F or Both	Acc	Operating Voltage Range	Alarm Signaling	Major Pins	Measurement Range
**LM35 [42]**	Analog	${}^{{}^{\circ}}$C	0.5 ${}^{{}^{\circ}}$C Acc guaranteeable at +25 ${}^{{}^{\circ}}$C	4 V to 30 V	N	VCC, VOUT, GND	Range is −55${}^{{}^{\circ}}$ to +150 ${}^{{}^{\circ}}$C
**DS18B20 [43]**	Digital	Both	±0.5 °C Acc from −10 ${}^{{}^{\circ}}$C to +85 ${}^{{}^{\circ}}$C	3.0 V to 5.5 V	Y	GND, DQ, VDD, NC	Range is 55 ${}^{{}^{\circ}}$C to +125 °C and 67 ${}^{{}^{\circ}}$F to +257 ${}^{{}^{\circ}}$F
**MCP9700 [44]**	Analog	${}^{{}^{\circ}}$C	±4 ${}^{{}^{\circ}}$C (max.), 0 ${}^{{}^{\circ}}$C to +70 ${}^{{}^{\circ}}$C	2.3 V to 5.5 V	N	Vout, Vcc, GND, NC	Range is −40 ${}^{{}^{\circ}}$C to +125 ${}^{{}^{\circ}}$C
**TMP100 [45]**	Digital	${}^{{}^{\circ}}$C	±1 ${}^{{}^{\circ}}$C (Typical) from −55 ${}^{{}^{\circ}}$C to 125 ${}^{{}^{\circ}}$C and ±2 ${}^{{}^{\circ}}$C (Max) from −55 ${}^{{}^{\circ}}$C to 125 ${}^{{}^{\circ}}$C	2.7 V to 5.5 V	Y	ADD0, ADD1, ALERT, GND, SCL, SDA, V+	Range is −55 and +125 ${}^{{}^{\circ}}$C

Acc—accuracy; ADD—address select; °C/°F—centigrade/fahrenheit; DQ—data in/out; GND—ground; N—no; NC—no connection; SDA—serial data; SDN—shutdown control; V—voltage; VCC—voltage common collector; VDD—power supply voltage; VOUT—output; Voltage Y—yes.

**Table 7 sensors-22-08073-t007:** Features of blood pressure sensors/devices.

BP Sensors	Features									
	Freq	Range	Major Pins	Pressure Hysteresis	Lin	Supply Voltage	Full Scale Span	RT	Offset Stability	Acc
**MPX10** **Series** **Pressure** **Sensor** **[46]**	N/A	0–10 kPa	GND, Vs, +Vout, −Vout	±0.1 typical	Min −1.0, Max 1.0	3.0–6.0 Vs	Min 20 mV, Max 50 mV	1.0 ms	±0.5% VFSS	N/A
**Omron** **HBP-** **1300** **digital** **Device** **[47,48]**	50/60 Hz	0 to 300 mmHg	Start/Stop, Mode, Last Reading (buttons)	N/A	N/A	100–240 V AC	N/A	N/A	N/A	Within ±3 mmHg
**Typical** **BP** **Monitor** **Sensor** **[49]**	N/A	0 to 258 mmHg	Tube, Pressure Cuff, Pressure Control Valve, Bulb	typical ±0.25%	typical ±0.25%	N/A	N/A	1.0 ms	N/A	±1 mmHg

Acc—accuracy; BP—blood pressure IR; kPa—kiloPascal’s; LED—infrared light-emitting diode; mmHg—millimeters of mercury; ms—millisecond; N/A—not applicable; RT—response time; V—voltage; Vout—voltage output; Vs—power supply.

**Table 8 sensors-22-08073-t008:** Features of oxygen sensors/devices.

Oxygen Sensors (Oximeter)	Features								
	INT	Addressed Parameters	Power Supply Voltage	Type	Acc SpO2	Acc PR	Major Pins	SpO2 Range	PR Range
**MAX30100 [36]**	Y	HR, SpO2	1.8 V to 3.3 V	IR LED	99.62%	97.55%	VIN, SCL, SDA, interrupt, IRD, RD, GND	N/A	N/A
**SpO2 Sensor** **Device [50]**	Y	HR, SpO2	D.C. 3.4 V ∼D.C.4.3 V	IR LED	±2% (80–100%); ±3% (70–79%)	±2% bpm	N/A	35 to 100%	25 to 250 bpm

Acc—accuracy; GND—ground; INT—integrated; IRD—IR led to driver; HR—heart rate; N/A—not applicable; PR—pulse rate; RD—red LED to driver; SDA—serial data; SDN—shutdown control; V—voltage; VIN—voltage input; Y—yes.

**Table 9 sensors-22-08073-t009:** Features of ECG sensor/devices.

ECG Sensors/ Devices	Features										
	INT	Single/ Multi Lead	Low Supply Current	Elec	SR	Right Leg Drive Shut Down	Single Supply OPER	HPF	Out	OPER TEMP	Major Pins
**AD8232 [32]**	Y	Single Lead	170 µA (typical)	2 or 3	360 HZ	N	2.0 V to 3.5 V	2 Poles	Rail to Rail	40 °C to +85 °C	GND, 3.3 V, OUT, LO−, LO+, ∼SDN, RA, LA, RL
**Holter** **Device [34]**	Y	Multi Lead	N/A	3, 5 or 12	125 HZ	N/A	one AAA battery	N/A	ECG Signal Rec on Monitor	+10 °C to +40 °C	Multiple Leads
**ADAS1000 [51]**	Y	Multi Lead	N/A	5 or 6	800 HZ	N/A	3.15 V to 5.5 V	N/A	Monitor	−40 °C to +85 °C	64 lead LQFP [52], 56 lead LFCSP (Both has diff. pins)
**AD8233 [53]**	Y	Single Lead	50 A typical	2 or 3	N/A	Y	1.7 V to 3.5 V	2 Poles Adjustable HPF	Rail to Rail	−40 °C to +85 °C	20 pins (GND, VS+, REFIN, HP- SENSE, HP- DRIVE, SDN, AC/ DC, FR, etc.
**Shimmer 3** ** [54,55,56,57]**	Y	Multi Lead	T60 µA Maximum	4	24 MHZ	N/A	450 mAh battery	N/A	On Windows PC and SQL	N/A	5 ECG pins, 5 EMG pins

AC/DC—alternating current/direct current; EMG—electromyography; FR—fast restore; GND—ground; HPF—high pass filter; HPSENSE—high pass sense; HPDRIVE—high-pass driver; HZ—hertz; INT—integrated; LA—left arm; LO—leads off; MHZ—megahertz; NA—not applicable; N—no; OPER TEMP—operation temperature; PC—personal computer; RA—right arm; REFIN—reference buffer input; RL—right leg; SDN—shutdown control input; SQL—structure query language sampling rate; V—voltage; Vs—power supply terminal; Y—yes.

**Table 10 sensors-22-08073-t010:** Sensors used in previous studies.

Year	Sensors Used						
	ECG Module	Temp Sensor	BP Sensor	Pulse/ HB Sensor	Oxygen Sensor	Other Sensor /Device	Integrated Sensor
**2016 [17]**	✗	MCP9700	✗	Pulse Sensor	✗	✗	✗
**2016 [58]**	✗	✗	✗	✗	✗	PCG Sensor	✗
**2016 [59]**	✗	✗	✗	✗	✗	✗	Wrist band for HB & BP (DNNS)
**2016 [19]**	✗	✗	✗	FingerTip- Optical Sensor for PPG	✗	✗	✗
**2016 [60]**	✗	✗	✗	✗	✗	Wearable Watch (PPG sensor)	✗
**2016 [61]**	Galilio Board plateform for ECG (UB-MMNS)	✗	✗	✗	✗	✗	✗
**2017 [37]**	Holter Devices		(UB-MNNS)	Holter Device, SpO2 Device (UB-MNNS)	SpO2 Sensor Device (DNNS)	✗	
**2017 [62]**	✗	✗	✗	Pulse Sensor	✗	✗	✗
**2017 [19]**	✗	18DS20	✗	KY-093	✗	✗	✗
**2017 [29]**	✗	✗	OMRONH- -BP1300	PPG Sensor		External Defibillator	✗
**2017 [63]**	(UB-MNNS)	(UB-MNNS)	(UB–MNNS)	HB Sensor	✗	Alchol Sensor, EMG (MNNS)	✗
**2017 [64]**	Wearable SOC ECG (MNNS)	✗	✗	✗	✗	✗	✗
**2018 [65]**	✗	✗	✗	✗	✗	✗	MAX30100 (SpO2, HB)
**2018 [66]**	✗	✗	✗	Pulse Sensor	✗	✗	✗
**2018 [18]**	✗	✗	✗	Pulse Sensor	✗	✗	✗
**2018 [38]**	ECG Module AD8232	✗	✗	Pulse Sensor	✗	✗	✗
**2018 [20]**	ECG Module AD8232	✗	✗	MAX30100	MAX30100	✗	✗
**2018 [24]**	✗	LM35	(UB–MNNS)	HB Sensor	✗	✗	✗
**2018 [67]**	✗	✗	✗	✗	✗	✗	WWSN for ECG, BP, Respiratory
**2019 [28]**	✗	18DS20	✗	HB sensor	✗	✗	✗
**2019 [25]**	Pulse Sensor	LM35	✗	Pulse Sensor	✗	✗	✗
**2019 [26]**	✗	LM35	✗	Pulse Sensor	✗	✗	✗
**2019 [40]**	ECG Module AD8232	✗	✗	✗	✗	✗	✗
**2019 [21]**	ECG Module AD8232	✗	✗	ECG Module AD8232	✗	✗	✗
**2019 [68]**	✗	✗	✗	✗	✗	Bio Sensors of hospital	✗
**2019 [69]**	✗	✗	✗	✗	✗	✗	Watch for HB, CL, bp, (DNNS)
**2019 [30]**	ECG AD8232 Module	✗	BP Cuff (UB-MNNS)	Heart Rate Monitor	✗	Near Infrared Sensor for CL	✗
**2019 [70]**	ECG AD8232 Module	✗	✗	✗	✗	✗	✗
**2019 [71]**	(UB-MNNS)	✗	✗	✗	✗	✗	✗
**2020 [72]**	(UB-MNNS)	✗	✗	✗	✗	✗	✗
**2020 [73]**	✗	(UB-MNNS)	(UB-MNNS)	Pulse Sensor	✗	✗	✗
**2020 [74]**	✗	✗	✗	HB sensor	✗	Alchohal Sensor (MNNS)	✗
**2020 [75]**	✗	✗	✗	✗	✗	✗	MD, AC, ENV Sensors (MNNS)
**2020 [27]**	ECG Module AD8232	LM35	MPX10	Pulse Sensor	Pulse Sensor	✗	✗
**2020 [76]**	3 Lead VCG signals (MNNS)	✗	✗	✗	✗	✗	✗
**2020 [22]**	ECG Module AD8232	✗	✗	ECG Module AD8232	✗	✗	✗
**2020 [77]**	ADAS1000	TMP100	✗	✗	✗	✗	✗
**2020 [78]**	Multiple ECG devices (MNNS)	✗	✗	✗	✗	✗	✗
**2020 [79]**	Shimmer3 ECG Unit	✗	✗	✗	✗	✗	✗
**2020 [23]**	ECG Module AD8232	✗	✗	ECG Module AD8232	✗	✗	✗
**2021 [80]**	✗	✗	(UB-MNNS)	(UB-MNNS)	✗	Glucose Sensor (MNNS)	✗
**2021 [81]**	Wearabale Smart ECG device (UB-DNNS)	✗	✗	✗	✗	✗	✗
**2021 [82]**	Self Made Device for ECG (NNS)	✗	✗	✗	✗	✗	✗
**2021 [83]**	Ready made ECG Device (UB-DNNS)	✗	✗	✗	✗	AllCheck Device	✗
**2021 [84]**	Multiscale ECG from 3 Sensors (UB-MNNS)	✗	✗	Wearable HB Sensor (MNNS)	Respiratory Sensor (MNNS)	Optical Sensor (MNNS)	✗
**2021 [41]**	ECG Module AD8283	✗	✗	✗	✗	✗	✗

BP/bp—blood pressure; CL—cholesterol; DNNS—device name not specified; ECG—electrocardiogram; HB—heartbeat; HR—heart rate; INT—integrated; MNNS—module number not specified; PCG—phonocardiograph; PR—pulse rate; PPG—photoplethysmography; UB-DNNS—used but device name not specified UB-MNNS—used but module number not specified; ✗—The specified parameter is not addressed.

### 5.2. RQ 2: What Are the Most Important Communication Technologies Used in E-Cardiac Care?

Communication technologies and protocols can be defined as a set of rules, technologies, semantics, equipment, and programs used to transfer, process, communicate, and receive information. Communication technologies and protocols vary depending upon the technology and network type devised, developed, or utilized. Some of the protocols and communication technologies are discussed in this section. The publications mentioned in Table 11 address the communication technologies and protocols used in previous selected studies for the development of E-Cardiology, monitoring, detection, and classification. BL is a wireless technology for short-range communication and exchanging data between mobile and fixed devices. BL has a transmission power of 1 mw–100 mw and a 1 Mbps data rate. Its data transmission range is 30 feet. The wearable healthcare monitoring devices (wearable fitness watches and pulse oximeters) may have the BL features integrated. BL technology was also employed in previous research [19,39,41,58,66,73] for data transmission for E-Cardiology. In prior literature on E-Cardiology monitoring, BL technology was determined to be the most commonly used technology. Ethernet is a wired communication networking protocol that can be used in local area networks (LANs), metropolitan area networks (MANs), and wide area networks (WANs). Ethernet allows communication through data cables. The publications [58,83] used an Ethernet-wired technology for connectivity support between various hardware modules implemented for cardiovascular disease diagnosis. One existing research study found that Ethernet-wired communication is rarely used in E-Cardiology. GSM is a cell-based or mobile communication modem that works as a mobile communication system. GSM technology is also used in E-Cardiology to send SMS messages or dial calls. GPS, which helps people to find their position on Earth, consists of networks of satellites and receivers or devices that determine location.

The communication technologies and protocols used in E-Cardiology in previous research studies and findings are detailed in Table 11.

### 5.3. RQ 3: Which Pre-Processing Techniques Are Used in E-Cardiology, along with the Most Widely Used AI Classifiers/Models?

RQ 3 is divided into two subsections. The first subsection investigates and compares various AI Models for the classification and prediction of CVD. This part explores various studies that use different machine learning and deep learning models for CVD prediction. Our study also provides a comprehensive explanation about the algorithms and methodologies used for prediction and classification techniques and the different datasets and performance metrics that we used to evaluate the models. Furthermore, the data preprocessing techniques used with different classifiers are also indicated in the second subsection below.

#### 5.3.1. AI Classifiers/Models and E-Cardiology

The prediction of CVD is a much discussed topic of research in the realm of healthcare. AI-based prediction systems can be of great help in detecting disease at an earlier stage which can reduce risk associated with disease progression. The concept of AI is not new in cardiac electrophysiology with automated ECG interpretation. It has existed in some form or other since the 1970s [87].

Artificial Intelligence (AI) is the reflection of human cognitive functions from the surroundings acquired by applying algorithms, pattern matching, cognitive computing, and deep learning to achieve specific objectives [88]. The ongoing progress in AI, primarily in the sub-domains of machine learning (ML) and deep learning (DL), have caught the attention of physicians hoping to develop newly integrated, dependable, and potent methods for ensuring standard healthcare in the critical field of cardiology.

Machine learning (ML) is a subset of AI to “teach” computers to analyze huge datasets in a quick, accurate, and efficient manner by using complex computing and statistical algorithms [89]. Supervised ML is more successful in predicting survival compared to the traditional clinical risk scores [90].

The study [91] proved that the accuracy of disease prediction can be increased by using an unsupervised type of ML for obstructive coronary artery disease in nuclear cardiology.

Deep learning (DL) is a supervised ML methodology that relies on neural networks and is known for the automated algorithms required to extract meaningful patterns from data collections [92]. In the medical context, the most widespread deep learning algorithms are artificial neural networks (ANN), multilayer perceptron (MLP), convolution neural networks (CNN/ConvNet), recurrent neural networks (RNN), radial basis function network (RBFN), deep belief networks, and deep neural networks (DNN) [88]. Compared with traditional supervised ML, the real strength of DL is that it is an effective, powerful, and flexible approach to representing complicated raw input data that does not demand manual feature engineering. For instance, while addressing the issue of automated ECG interpretation, early conventional supervised ML techniques depended on human-defined ECG features. In contrast, the modern DL model extracts patterns within raw ECGs to detect sinus rhythm and various other arrhythmias with a performance that equals the result of any cardiologist [93].

The significant areas of cardiac healthcare that can benefit from ML/DL techniques are prognosis, diagnosis, classification, treatment, and clinical workflow. Table 12 presents an overview of different AI algorithms extracted from the literature review on heart disease diagnosis/classification.

A comparative analysis of different AI techniques frequently used in Smart cardiology for the prognosis/diagnosis of various CVDs is given in Table 13.

As suggested by WHO, by 2030 almost 23.6 million individuals will die from heart-related causes [94]. CVDs are the main cause, but they can be cured and prevented. To reduce the risk involved, analysis is fundamental. The difficult part is accurate diagnosis [95].

Table 14 summarises the most recent work performed in the field of artificial intelligence related to CVDs.

#### 5.3.2. Data Preprocessing Techniques in E-Cardiology

This section identifies and evaluates studies that applied data preprocessing techniques in cardiac disease classification. Data Preprocessing (DP) in AI is a critical stage that enhances the quality of data to achieve meaningful insights and is the initial step in the development of an AI model. Conventionally, real-world data is not in an appropriate format and contains errors or outliers. It usually lacks specific attribute values/trends, thus resulting in an inadequate AI model. Data preprocessing solves this problem by cleaning and organizing raw data to tailor it to the needs of building and training AI models. Hence, data preprocessing in AI is a data mining approach that reshapes raw data into a readable format that is readily available for an AI model to meet the high standards of performance [120]. Consequently, the algorithm can easily interpret the data’s features.There are four primary ways of data preprocessing, i.e., (1) data cleaning, (2) data integration and formatting, (3) data transformation, and (4) data reduction.

Different preporocessing techniques used in past studies for diagnosing heart disease and other types of arrhythmia are also mentioned in Table 14, along with AI models. This table also lists the task performed by the preprocessing technique.

### 5.4. RQ 4: What Are the Major Issues and Challenges in Current E-Cardiology?

After conducting comprehensive research, we identified some significant benefits and major challenges in the field of MIoT to answer our RQ 4. These challenges and benefits have been emphasized on the basis of studies conducted by different researchers in the domain of MIoT and E-Cardiology. Based upon selection and rejection criteria, only valid and reliable papers were selected, as mentioned earlier. We incorporated only the latest benefits and challenges that were found to be unique in the domain of IoT and AI regarding healthcare and cardiology. A pictorial representation of these benefits and challenges is shown in Figure 4. 

#### 5.4.1. Benefits of E-Cardiology

***Internet of Things (IoT)*** develops a linkage between “things”, such as devices, gadgets, vehicles, and sensors. Likewise, the medical Internet of Things (MIoT)-based cardiovascular healthcare system monitors the physical symptoms [5] of cardiac patients at a very reasonable cost. These physical symptoms include temperature, blood pressure (BP), SPO2, and heartbeat, along with ECG [6] and associated numerical measurements. Significant benefits of IoT-based cardiology from various studies are noted in Table 15.

A comparison of some key factors in Smart cardiac care are shown in Table 16.

***Artificial intelligence (AI)*** is another significant aspect of E-Cardiology. Until now, AI in cardiac electrophysiology has exhibited promising results. Primary advantages of AI-based cardiology are discussed in Table 17.

#### 5.4.2. Challenges of E-Cardiology

***Internet of Things (IoT)*** comes with various challenges as shown in Table 15. In addition to these challenges, ref. [129] proposed some other vital challenges, such as fixation of sensors, body impact on signal propagation, and synchronization, that may affect critical health services such as cardiac care. The MIoT-based healthcare systems are expected to produce a vast amount of data. Moreover, these sensors and devices are linked through networks, thus enabling real-time transmission of data. Therefore, hackers may attempt to target it. Moreover, the timely availability of medical data will affect the patient’s life. Consequently, it is crucial to have real-time information with lower latency over the network [124].

***Artificial intelligence (AI)*** has brought a revolution in the field of healthcare. It has especially made a great contribution in the domain of cardiac care, such as timely prediction and diagnosis of cardiovascular diseases (CVDs), ECG analysis, and arrhythmia classification. However, despite all these milestones, it carries some challenges. Table 17 describes critical issues and challenges of AI-based cardiology from different past studies.

It is hoped that these challenges can be met and that through MIoT and AI we can achieve new levels of technical and medical standards in the field of cardiac healthcare.

## 6. Discussion

Figure 5 shows the distribution of the selected studies (104 papers) according to our four research questions (RQs). The pie chart shows that 47 studies overlap RQ1 and RQ2, whereas 33 studies address RQ3 and 24 address RQ4.

The outcomes of this systematic literature review suggest a noticeable increase in the research conducted in IoT using artificial intelligence in E-Cardiology. The research activities also incorporate monitoring CCU parameters, ECG analysis, and diagnosis/classification of various heart diseases. This extensive review study reveals that IoT sensors utilized for E-Cardiology are based upon analog sensors, digital sensors, and different wearable sensors or device modules. For fitness tracking and health monitoring activities, a variety of ready-made and wearable watches are also available. E-Cardiology patients can use a variety of wearable gadgets to track various cardiac healthcare characteristics. To establish expert E-Cardiology systems, several communication technologies and protocols for transmissions over a defined range need to be instituted and maintained. Our findings on communication technologies and protocols for the implementation of IoT-based cardiovascular healthcare monitoring and detection systems show that the following technologies provide viable tools to use in MIoT systems: Bluetooth (BL/BLE), ethernet, global system for mobile (GSM), global positioning system (GPS), global packet radio service (GPRS), message queuing telemetry transport (MQTT), short message service (SMS), email, zigbee, transmission control protocol/Internet protocol (TCP/IP), wireless fidelity (Wifi), security protocols, broadband/Internet, cloud, Smart cars, Smart phones/computers, etc. Our findings show that deep learning is often used in cardiac imaging procedures, particularly in echocardiography [88]. Furthermore, CNNs have been evaluated and found useful in calculating coronary artery calcium in cardiac CT angiography [101]. Though deep learning techniques are garnering attention in the field of Smart cardiology, we may infer that instead of depending on a particular AI model, hybrid techniques are expected to produce better results. From the results of our study, it may also be inferred that SVM was more frequently used in cardiac care than was deep learning. However, in recent years, deep learning has emerged as a more powerful and reliable tool for the detection and diagnosis of various heart diseases. It was also noted that data reduction appears to be a major concern of researchers when applying data mining/data preprocessing approaches to predict CVD. This literature review includes 24 studies devoted to major issues and challenges in E-Cardiology. These studies conclude that a major benefit of MIoT in cardiac healthcare is that it generates timely and accurate data, which results in better healthcare outcomes. One vital advantage of using ML techniques is the ability to fuse various types of data [143]. We also assessed the results and noted that mIoT devices do not possess the requisite data protocols and standards [131]. Therefore, many issues must be addressed to ensure IoT privacy and security [144,145], which is one of the major challenges of the IoT era. Moreover, the continuous monitoring of critical indicators requires reduced energy consumption and a longer battery life [146] to prevent a break of communication. This is also one of the significant challenges of MIoT. The medical data gathered using MIoT is seldom standardized and often fragmented. The data in legacy IT systems are usually generated with incompatible formats. Thus, the great challenge of interoperability needs to be addressed as well [124,147]. However, to move forward in the field of MIoT, fearing AI is no option.

Instead, we should work toward the smooth digitization of healthcare infrastructure [148]. Obviously, various benefits of AI cannot be implemented and utilized correctly without integrating AI into clinical decision making effectively and responsibly [149].

Figure 6 refers to the research work conducted based on the number of papers per annum. It shows the year-wise trends in publications in the field of E-Cardiology. It also indicates that the maximum number of papers selected for this survey was from 2019.

### 6.1. Gaps, Future Recommendations

In a future work, all the vital cardiac parameters can be combined with an intelligent cardiac care unit (CCU) to develop a complete picture of E-Cardiology. These parameters/indicators may be comprised of temperature, blood pressure, oxygen saturation, heart rate, and ECG analysis. In addition, in differentiating between normal and abnormal heartbeats, a Smart CCU can also be used to detect QRS complexes in electrocardiographic (ECG) signals to determine the presence of a cardiac malady and different arrhythmias. Integrated and wearable IoT solutions, which address all the necessary cardiac parameters of a heart patient, need to be implemented. The results and accuracy of the devices/sensors used in the development of an IOT-based cardiac system cannot be compromised. The implemented system must be tested, evaluated, and approved under the supervision of cardiologists. Advanced communication technologies, including secure network protocols, must be implemented. Data accessibility features, such as widespread data access, must be possible in a secure environment so that the data confidentiality and integrity are maintained. Ubiquitous access is also an important factor and can be achieved by storing the digital data on a cloud server.

### 6.2. Limitations of the Review Study

This review of literature has some limitations. First, many papers were conference proceedings; therefore some parameters remained inaccessible since their authors did not mention them in detail. Second, some of the studies on AI-based IoT architecture in cardiac healthcare could not be located even after following a comprehensive search protocol, such as gray literature and reports that were not published in the databases which we selected for review. Therefore, we suggest that an additional systematic study be conducted to cover the related literature from other important databases.

## 7. Conclusions

This review study outlines a total of 104 primary studies from 2016 to 2021 based on our filtering process for supporting the proposed research. Quality assessment of the selected studies was conducted for the formulated research questions after a rigorous analysis.

This work mentions different sensors and communication technologies being used in cardiology. Moreover, this review study also describes various preprocessing techniques and AI algorithms used in the existing studies to diagnose and classify CVD and ECG analysis. This systematic review also provides comparative analysis of various existing techniques in the field of AI, medical sensors, and communication technologies. Finally, this study targets various advantages and issues indicated in the existing literature in the field of E-Cardiology. The interaction of MIoT and artificial intelligence makes cardiac healthcare more manageable by making various applications, services, communication protocols, third party APIs, and IoT sensors available. E-Cardiology guarantees more privacy and security to the IoT devices which are prone to hackers. Furthermore, AI-based diagnosis of various cardiovascular diseases in E-Cardiology helps save time, enabling cardiologists to focus more on treatment. This systematic work presents a review protocol to analyse how IoT applications assist cardiac healthcare and how various artificial intelligence (AI) models contribute to present and prospective research work of IoT in E-Cardiology. This study also indicates how different communication technologies bring privacy and security to IoT devices related to cardiac healthcare. The purpose of this paper is to highlight the influence of IoT, communication technologies, and AI techniques in cardiac healthcare in light of our systematic literature review protocol. Therefore, one can say that this systematic review covers the complete and latest infrastructure of E-Cardiology, along with its benefits and challenges which have not been examined before in such a comprehensive way.

## Figures and Tables

**Figure 1 sensors-22-08073-f001:**
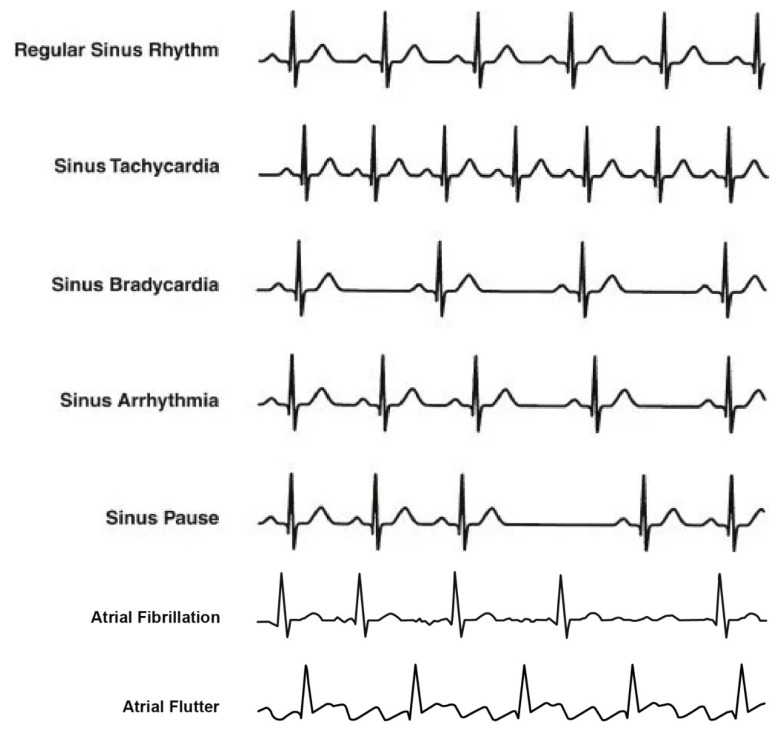
Types of arrhythmia.

**Figure 2 sensors-22-08073-f002:**
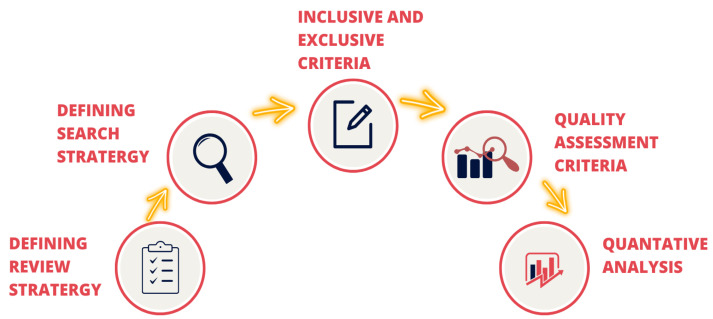
SLR protocol outline.

**Figure 3 sensors-22-08073-f003:**
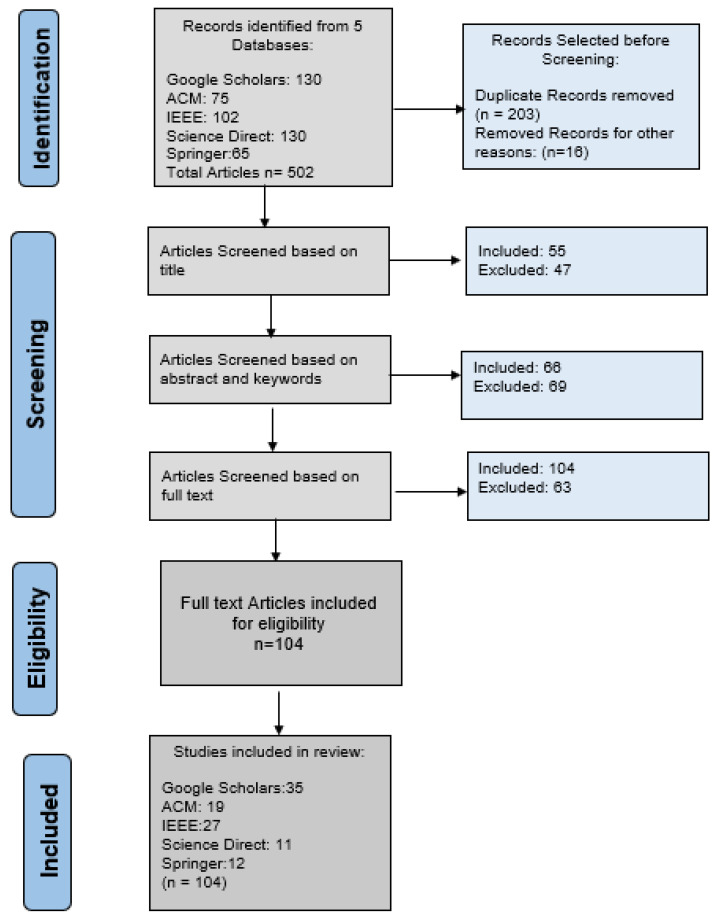
Flow diagram showing the screening process for the systematic review.

**Figure 4 sensors-22-08073-f004:**
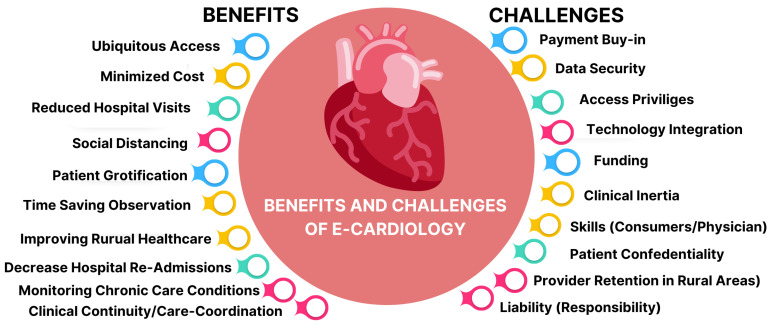
Benefits and challenges of E-Cardiology.

**Figure 5 sensors-22-08073-f005:**
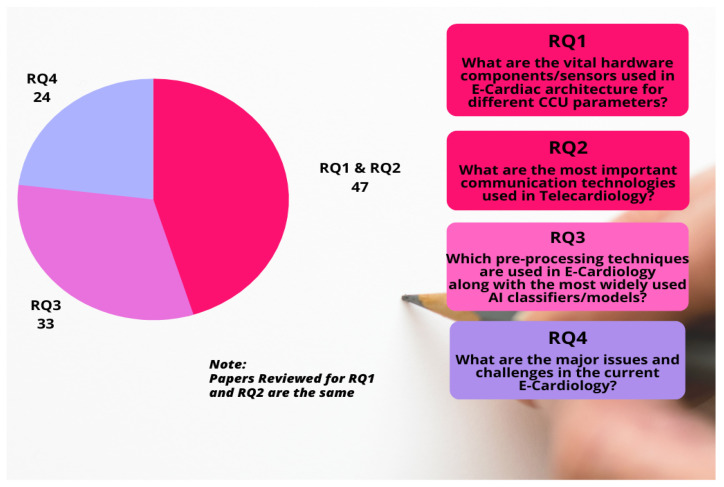
Statistical analysis of reviewed papers in terms of formulated RQs.

**Figure 6 sensors-22-08073-f006:**
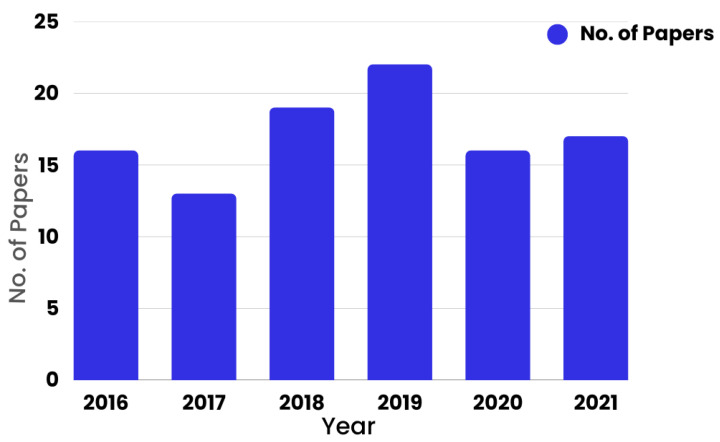
Year-wise trends in publications in the field of E-Cardiology.

**Table 1 sensors-22-08073-t001:** Various Types of Arrhythmias.

Types of Arrhythmia	Explanation
Tachyarrhythmias	A fast heart rhythm with a rate of more than 100 beats per minute.
Bradyarrhythmias	Slow heart rhythms that may be caused by disease in the heart’s conduction system.
Supraventricular arrhythmias	Arrhythmias that begin in the atria (the heart’s upper chambers). “Supra” means above; “ventricular” refers to the lower chambers of the heart, or ventricles.
Ventricular arrhythmias	Arrhythmias that begin in the ventricles (the heart’s lower chambers).

**Table 2 sensors-22-08073-t002:** Review questions and their motivation.

No.	Review Question	Motivation
RQ 1	What are the vital hardware components/sensors used in E-Cardiac architecture for different CCU parameters?	The main focus of this question is to identify different types of sensors and their features most often used in IoT-based Cardiac Healthcare.
RQ 2	What are the most important communication technologies used in E-Cardiac Care?	The question aims to find the most commonly used communication technologies in MIoT-based Cardiology.
RQ 3	Which pre-processing techniques are used in E-Cardiology along with the most widely used AI classifiers/models?	This question is designed to explore the current work in the field of medical IoT accompanied by artificial intelligence in cardiac healthcare and to identify various classification and preprocessing techniques used for predicting cardiovascular diseases.
RQ 4	What are the significant issues and challenges in the current E-Cardiology?	This question investigates major benefits, and current challenges of IoT-based cardiac healthcare system.

**Table 3 sensors-22-08073-t003:** Search strings used for data retrieval.

No.	Search String
1	(internet of things OR IoT OR IoT-based OR smart health) AND (cardiac OR heart OR CCU) AND (monitoring OR detection OR diagnosis OR disease OR parameters)
2	(intelligent OR artificial intelligence OR AI OR machine learning OR deep learning OR preprocessing OR reduction OR cleaning OR data mining) AND (cardiac OR heart OR ECG OR arrhythmia OR cardiovascular OR smart health OR healthcare OR smart healthcare OR cardiology) AND (technique OR methods OR classification OR algorithm)
3	(internet of things OR IoT OR IoT-based) AND (cardiac OR heart OR CCU OR smart health OR smart healthcare) AND (benefits OR advantages OR challenges OR issues OR disadvantages)

**Table 4 sensors-22-08073-t004:** Inclusion and exclusion criteria.

Inclusion Criteria	Exclusion Criteria
The papers published in English were chosen on priority.	The papers published in other languages were not selected.
The most recently published research papers, i.e., 2016 to 2021, were singled out for studies.	Gray literature was excluded from the study list.
Papers describing an overview of current approaches that implement modern tools and techniques in E-Cardiology were selected.	Papers not defining the topic appropriately were excluded.
The main aim was to target the primary studies such as original research papers.	Duplicated material was removed.

**Table 11 sensors-22-08073-t011:** Communication technologies used in previous studies.

Year	Communication Technologies Used														
	**BT**	**ETH**	**GSM**	**GPS**	**GPRS**	**MQTT**	**SMS**	**SMPP**	**ZB**	**WIFI**	**TCP/** **IP**	**Sc.** **P**	**Cloud**	**SP/** **PC**	**Internet**
**2016 [17]**	N	N	Y	Y	Y	N	Y	Y	N	Y	N	N	Y	Y	N
**2016 [58]**	Y	Y	N	N	N	N	N	N	N	Y	Y	Y	Y	Y	N
**2016 [59]**	Y	N	N	N	N	N	Y	N	N	Y	N	Y	Y	N	N
**2016 [85]**	N	N	N	N	N	N	Y	N	N	N	N	N	N	Y	N
**2016 [60]**	N	N	N	N	N	N	N	N	N	N	N	N	N	Y	N
**2016 [61]**	N	N	N	N	N	N	N	N	N	Y	N	N	N	Y	N
**2017 [37]**	Y	N	Y	N	Y	N	N	N	N	Y	N	N	N	Y	Y
**2017 [62]**	N	N	N	N	N	N	N	N	N	N	N	N	N	N	N
**2017 [19]**	Y	N	Y	N	N	N	Y	N	N	Y	Y	N	Y	Y	N
**2017 [29]**	N	N	Y	N	Y	N	N	N	N	Y	N	N	Y	Y	N
**2017 [63]**	N	N	N	N	N	N	N	N	N	Y	Y	N	Y	Y	N
**2017 [64]**	N	N	N	N	N	N	N	N	N	Y	N	Y	Y	Y	N
**2017 [39]**	Y	N	N	Y	N	N	N	N	N	Y	N	N	N	Y	N
**2018 [65]**	N	N	Y	Y	Y	N	Y	N	N	Y	N	N	Y	Y	N
**2018 [66]**	Y	N	N	N	N	N	Y	N	N	Y	N	Y	Y	Y	N
**2018 [18]**	N	N	N	N	N	N	N	N	N	Y	N	N	Y	Y	N
**2018 [38]**	N	N	Y	N	N	N	Y	N	N	Y	Y	N	Y	Y	N
**2018 [20]**	N	N	N	N	N	N	N	N	N	Y	Y	N	Y	Y	N
**2018 [24]**	N	N	N	N	N	N	N	N	N	Y	Y	N	N	N	N
**2018 [67]**	Y	N	Y	Y	Y	N	N	N	N	Y	N	N	N	Y	N
**2019 [28]**	N	N	N	N	N	N	Y	N	N	N	N	N	N	Y	N
**2019 [25]**	Y	N	N	N	N	N	N	N	N	N	N	N	N	Y	N
**2019 [26]**	N	N	N	N	N	N	N	N	N	Y	Y	Y	Y	N	N
**2019 [40]**	N	N	N	N	N	N	N	N	N	Y	N	Y	Y	Y	N
**2019 [21]**	Y	N	N	Y	N	N	N	N	N	N	N	N	N	Y	N
**2019 [68]**	N	N	N	N	N	N	N	N	N	Y	N	Y	Y	Y	N
**2019 [69]**	Y	N	N	N	N	N	N	N	N	Y	N	N	Y	Y	N
**2019 [30]**	Y	N	N	N	N	N	N	N	N	Y	N	Y	Y	Y	N
**2019 [70]**	N	N	N	N	N	N	Y	N	N	N	N	N	N	Y	N
**2019 [71]**	Y	N	N	N	N	N	N	N	N	N	N	N	Y	Y	N
**2020 [72]**	N	N	N	N	N	N	N	N	N	Y	N	N	N	N	N
**2020 [86]**	N	N	N	N	N	N	N	N	N	Y	N	N	N	Y	N
**2020 [73]**	Y	N	Y	Y	Y	N	N	N	N	Y	N	N	Y	Y	N
**2020 [74]**	N	N	Y	N	Y	N	Y	N	Y	N	N	N	N	N	N
**2020 [75]**	Y	N	N	N	N	N	N	N	N	Y	Y	Y	Y	Y	N
**2020 [27]**	N	N	N	N	N	N	N	N	Y	Y	N	N	N	Y	N
**2020 [76]**	N	N	N	N	N	N	N	N	N	N	N	N	N	N	N
**2020 [22]**	N	N	N	N	N	Y	N	N	N	Y	N	N	Y	Y	N
**2020 [77]**	Y	N	Y	N	Y	N	Y	N	N	Y	Y	Y	Y	Y	N
**2020 [78]**	N	N	N	N	N	N	N	N	N	Y	N	N	Y	Y	N
**2020 [79]**	N	N	N	N	N	N	N	N	N	Y	N	N	Y	Y	N
**2020 [23]**	Y	N	N	N	N	N	N	N	N	Y	Y	Y	Y	Y	N
**2021 [80]**	Y	N	N	N	N	N	N	N	N	Y	N	Y	Y	Y	N
**2021 [81]**	N	N	N	Y	N	N	N	N	N	N	N	N	N	N	N
**2021 [82]**	Y	N	N	N	N	N	N	N	N	N	Y	N	Y	Y	N
**2021 [83]**	N	Y	N	N	N	Y	N	N	N	Y	Y	Y	Y	Y	N
**2021 [84]**	N	N	N	Y	N	N	N	N	N	Y	N	N	N	N	N
**2021 [41]**	Y	N	N	N	N	N	N	N	N	Y	N	N	Y	Y	N

BT—Bluetooth; GSM—global system for mobile communications; GPS—global positioning system; GPRS—general packet radio service; MQTT—message queuing telemetry transport; N—No; Sc.P—security protocols; SMS—short message service; SMPP—short message peer-to-peer; SP/PC—Smart phone/personal computer; TCP/IP—transmission control protocol/Internet protocol; WIFI—wireless fidelity; Y—yes.

**Table 12 sensors-22-08073-t012:** Overview of common ai techniques used in E-Cardiology.

No.	AI Algorithm	Description	Strengths	Limitations
1	Principal Component Analysis (Unsupervised)	A method of dimensionality reduction which aims to compute principal components and makes data more compressible.	1. Compute principal components 2. Avoids data overfitting 3. High variance, improved visualization 4. Reduce Complexity	1. Low interpretability of principal components. 2. Dimensionality reduction may result in information loss.
2	K-Means Clustering (Unsupervised)	Generates k number of centroids that help to define clusters of data.	1. Ensures convergence 2. Can warm-start the positions of centroids 3. Easily adjusts to new examples 4. Assists the doctors in making more accurate diagnosis	1. Not suitable for data varying in size and density 2. Noise sensitive
3	Decision Tree (Supervised)	For classifying examples, a decision tree is an easy and simple representation.	1. Easy to interpret 2. Avoids over-fitting by pruning 3. Less sensitive to outliers 4. Requires less data cleaning	1. Instability 2. Relatively inaccurate
4	K-Nearest Neighbor (Supervised)	Saves all available cases and allocates new cases based on a similarity measure.	1. Easy to implement & understand 2. Used for both classification & regression problems	1. Significantly slow as the data size increase 2. Computationally expensive 3. Requires high memory
5	Naïve Bayes (Supervised)	An easy probabilistic classifiers based on Bayes’ theorem.	1. Scalable 2. Fast 3. Used for real-time predictions 4. Not requires large amounts of data	1. Assumes attributes are mutually independent 2. Zero Frequency limitation
6	Random Forest (Supervised)	A set of decision trees, usually trained with the “bagging” technique. It performs classification as well as regression tasks.	1. Used for prediction 2. Resistant to noise and overfitting 3. Flexible, can handle large datasets easily	1. Can take up lots of memory 2. Not that interpretable
7	Support Vector Machine (Supervised)	Indicates hyperplane which separates classes, based on a similarity measure, can be used as a linear or nonlinear kernel.	1. Fast 2. Relatively memory efficient 3. Works well with clear margin of separation between classes	1. Difficult to interpret 2. Not suitable for large datasets 3. May need normalization & scaling
8	Logistic Regression (Supervised)	The logistic paradigm can be used to model the probability of a certain class or event happening.	1. Easy to implement and interpret 2. Efficient to train	1. Performs poorly with large no. of variables 2. Used to predict only discrete functions 3. Not capture interactions auto- matically
9	Backpropagation (Supervised)	Backpropagation is a widely used algorithm for training feedforward neural networks. It is a reliable tool for increasing the accuracy of predictions.	1. Fast 2. Simple 3. Easy to analyze 4. Flexible	1. Sensitive to noisy/complex data 2. Performance of backpropagation depends on input data
10	Deep Learning (ANN) (Supervised)	Multilayered processing technique that mimics human neuronal structure. Different types of ANN are CNN or ConvNet, MLP, RBFN, RNN, etc.	1. No feature engineering 2. Learn complex functions 3. Enhanced Accuracy 4. Scalabale Model	1. Requires extremely large datasets 2. Intensive computational power 3. Difficult to interpret 4. Significant processing time

**Table 13 sensors-22-08073-t013:** Comparative analysis of various AI techniques used for prognosis/diagnosis of CVDs.

No.	AI Techniques Used in Smart Cardiology	Performance Metrics									
		Cl. Acc	Sensitivity	Specificity	Flexibility	Efficiency	Com. CPLX	Interpretability	Large Dataset Handling	Training Time	Noise Tolerance
1	DT	L	Y	Y	Y	Y	L	H	N	S	N
2	RF	H	-	-	Y	-	L	L	Y	F	N
3	NB	L	Y	-	Y	Y	L	H	Y	F	Y
4	PCA-KNN	H	Y	Y	-	-	-	L	-	F	N
5	SVM	H	Y	Y	Y	Y	H	L	N	S	N
6	LR	L	-	-	Y	Y	L	H	Y	F	N
7	BP	H	Y	Y	Y	-	-	H	Y	-	N
8	DL (ANN)	H	Y	Y	Y	Y	H	L	Y	S	Y

PCA-KNN—principal component analysis with K-nearest neighbor; NB—naïve Bayes; RF—random forest; SVM—support vector machine; LR—logistic regression; DL—deep learning; ANN—artificial neural networks; BP—backpropagation; Y—yes; S—slow; F—fast; L—low; H—high; N—no.

**Table 14 sensors-22-08073-t014:** Summary of AI-methodologies and data preprocessing techniques identified for E-Cardiology, from different studies.

Ref #	Year	Findings								
		AI Methodology	Prognosis/ Diagnosis Task	Types of CVDs	Cardiac Parameter/s	Cardiac Dataset	Preprocessing Task	Data Preprocessing Techniques	Accuracy %	Complexity
[94]	2016	DT	Coronary Heart Disease	N/A	N/A	UCI	N/A	N/A	86.7	M
[96]	2016	BBNN	Arrhythmia	5	ECG	MIT-BIH	Feature Extraction	Hermit Basis Function	97	H
[97]	2016	NN	Arrhythmia	5	ECG	MIT-BIH	Denoising Feature Extraction	DWT DWT + PCA DWT + ICA	98.91	H
[98]	2016	PSO tuned SVM	Arrhythmia	12	ECG	MIT-BIH	Feature Extraction	DOST	99.18	H
[99]	2016	NN	Arrhythmia	5	ECG	MITBIH	Feature Extraction	DOM	95	M
[100]	2016	RF	Arrhythmia	5	ECG	MIT-BIH	Feature Extraction	WPE	94.61	M
[61]	2016	SVM	Arrhythmia	2	ECG	CT	Feature Extraction	DWT	98.9	M
[101]	2016	Paired-CNN	Coronary Artery Calcification	N/A	CCTA	CT	Feature Extraction	ConvNet	Sens. = 71	H
[102]	2017	DL	Arrhythmia	N/A	ECG	MIT-BIH	Feature Extraction	AlexNet (DNN)	92	H
[103]	2017	SVM	Arrhythmia	4	ECG	MIT-BIH	Denoising Feature Extraction	Multiresolution DWT	98.39	M
[104]	2017	RBF-NN	Arrhythmia	6	ECG	MIT-BIH	Denoising Feature Extraction	DWT EMD Features	99.89	H
[102]	2017	DL	Arrhythmia	3	ECG	MIT-BIH	Feature Extraction	Transferred Deep Learning	92	H
[105]	2018	SVM	Arrhythmia	3	ECG	CUDB VFDB	Feature Extraction	FFT	95.9	M
[106]	2018	SVM	Arrhythmia	5	ECG	MIT-BIH	Denoising Normalization Feature Extraction	Daubechies wavelets min-max Normalization PCANet	97.77	M
[107]	2018	Twin LS-SVM	Arrhythmia	16	ECG	MIT-BIH	Feature Extraction	Composite Dictionary (DOST + DST + DCT)	99.21	H
[108]	2018	MPNN	Arrhythmia	3	ECG	MIT-BIH	Denoising Feature Extraction	Daubechies wavelet Multiresolution DWT MPNN	99.07	H
[109]	2018	DL	Arrhythmia	5	ECG	MIT-BIH	Normalization Feature Extraction	Z-score normalization CNN and LSTM	98.10	H
[110]	2018	DL	Arrhythmia	2	ECG	MIT-BIH	Feature Extraction	DNN	99.68	M
[111]	2018	DBN	Arrhythmia	5	ECG	MIT-BIH	N/A	N/A	95.57	H
[91]	2018	DNN	Arrhythmia	N/A	CCTA	CT	Feature Extraction	DNN	Sens. = 82.3	H
[71]	2019	CNN	Arrhythmia	4	ECG	CT MIT-BIH	Feature Extraction	CNN	94.96 95.73	M
[112]	2019	MPNN-BP	Heart Disease	5	ECG	UCI	N/A	N/A	97.5	H
[93]	2019	DNN	Arrhythmia	12	ECG	CT	Denoising Feature Extraction	DNN	ROC = 97 F1 = 83.7	H
[68]	2019	Hybrid Model	Heart Disease	8	ECG, HR, BP	CT	Data Cleaning Denoising Feature Selection	Numerical Cleaner Filter SFS	98	H
[113]	2020	CNN-KCL	Myocarditis	N/A	ECG	ZAS	Outlier Anomaly K-means clustering	K-means clustering CNN	92.3	H
[114]	2020	CNN	Myocardial Infarction	N/A	ECG	PTB	Data Augmentation Segmentation Feature Extraction	CNN	99.02	H
[115]	2020	DNN	Arrhythmia	6	ECG	TNMG	N/A	N/A	F1 = 80 Spec. = 99	H
[72]	2020	TWSVM	Arrhythmia	16	ECG	CT MIT-BIH	Feature Extraction	DWT	95.68	H
[78]	2020	DHCAF MCHCNN	Arrhythmia	5	ECG, HR	CT MIT-BIH	Denoising Feature Extraction	Daubechies wavelet-4 HWT	91.4 93	H
[116]	2021	E-D CNN-SVM	Myocardial Infarction	N/A	ECHO	HMC-QU	Featuring Engineering	CNN	80.24	H
[117]	2021	RF, NB	Coronary Heart Disease	N/A	N/A	OR	N/A	N/A	83.85 (RF) 82.35 (NB)	M
[118]	2021	AI	Cardiac Amyloidosis	N/A	ECG	MC	Feature Extraction	DNN	90	N/A
[119]	2021	CNN	Heart Failure	N/A	N/A	CT	Feature Selection	LASSO Regression	97	H

H—high; M—medium; N/A—not available; DL—deep learning; MPNN-BP—multilayer perceptron neural network-backpropagation; RF—random forest; WPE—wavelet packet entropy; CNN—convolutional neural network; CL—K means clustering; ZAS—Z alizadeh; PTB—physikalisch technische bundesanstalt; NB—naive Bayes; FFT—fast fourier transform; DNN—deep neural network; SVM—support vector machine; NN—neural networks; BBNN—block-based neural network; MC—Mayo Clinic; HWT—haar wavelet transform; EMD—empirical mode decomposition; RBNN—radial basis function neural network; OCAD—obstructive coronary artery disease; DBN—deep belief networks; ROC—receiver operating characteristic curve; DHCAF—dynamic heartbeat classification; SFS—sequential forward transform; DOST—discrete orthogonal stockwell transform; DOM—difference operation method; CCTA—cardiac CT angiography; PSO—particle swarm optimization; CUDB—Creighton University database; VFDB—ventricular fibrillation database; LS—SVM-least square SVM; OR—online repository; DBN—deep belief networks; ROC—receiver operating characteristic curve; DWT—discrete wavelet transform; DHCAF—dynamic heartbeat classification with adjusted features; MCHCNN—multi-channel heartbeat convolutional neural network.

**Table 15 sensors-22-08073-t015:** Key issues and major benefits of IoT-based cardiology.

Ref #	Year	Findings		
		Key Challenges and Barriers of E-Cardiology with IoT	Data Related Issues	Benefits of IoT-Based E-Cardiology
[121]	2016	Security, Interoperability Unintended Behavior, Device Vulnerability	Privacy Consistency Integration	Cost Reduction Clinical Continuity Quality Life, Telemedicine
[122]	2016	Security, Interoperability Complexity, Scalability Device Vulnerability	Privacy	Cost Reduction Clinical Continuity Automation, Time Saving
[123]	2017	Security Energy Consumption Network Latency Intelligence in Medical Care System Predictability	Privacy Real-Time Processing	Cost Reduction Clinical Continuity, Automation, Time Saving, Quality Life, Telemedicine
[124]	2018	Security, Interoperability Energy Consumption Network Latency	Privacy	Ubiquitous Access, Quality Life Cost Reduction, Time Saving Reduced Hospital Visits
[11]	2019	Security, Interoperability Energy Consumption Internet Bandwidth	Privacy	N/A
[125]	2019	Security Heterogeneity	Privacy, Reliability, Utility Validity, Generalizability Integrity, Objectivity, Data Overload, Completeness, Relevance	Personalized, Predictive, Participatory, Preventative, Persuasive, Perpetual, Programmable (7P)
[126]	2019	Security Unintended Behavior	Privacy, Confidentiality	Ubiquitous Access Cost Reduction Clinical Continuity Improved Accuracy Quality Life, Telemedicine
[127]	2019	Security Scalability	Privacy Data Overload	Ubiquitous Access Cost Reduction, Clinical Continuity Automation, Time Saving Quality Life Telemedicine
[128]	2019	Security Context-aware Computing Interoperability Energy Consumption	Privacy	N/A
[129]	2020	Unobtrusiveness Energy Consumption Quality of Service Scalabilty, Fixation Patient Indentification Body Impact on Signal Propagation	Reliability Integrity Data Protection Data Representation Accuracy	N/A
[130]	2020	Energy Consumption, Storage Patient’s discomfort caused by Sensors	Privacy Data Overload Noise	Flexibility Clinical Continuity Remote Monitoring
[131]	2020	Security	Privacy Confidentiality Integrity, Data Loss Aval1abilty Compromise	Remote Monitoring Cost Reduction, Time Saving Better Diagnostics Improved Clinical Infrastructure
[132]	2020	Security Mobility Heterogeneity Legal Aspects	Privacy	N/A
[133]	2021	Security, Scalability	N/A	Efficient, Cost Effective
[134]	2021	Security Scalability Interoperability, Energy Consumption, Low Latency Tolerance	Privacy Computational Intensity	Ubiquitous Access Time Saving, Cost Reduction Telemedicine, Quality Life Clinical Continuity Easy Usge

**Table 16 sensors-22-08073-t016:** Comparison of the existing evaluation factors in the Smart cardiac care.

Ref #	Year	Security	Privacy	Complexity	Integration	Reliability	System Predictability	Interoperability	Scalability	Heterogeneity	Energy Consumption	Network Latency
[122]	2016	✓	✓	✓	✓	✗	✗	✓	✗	✗	✗	✗
[121]	2016	✓	✓	✗	✓	✗	✗	✓	✗	✗	✗	✗
[135]	2017	✓	✗	✗	✗	✗	✓	✗	✗	✗	✓	✗
[123]	2017	✓	✓	✗	✗	✗	✓	✗	✗	✗	✓	✓
[124]	2018	✓	✓	✗	✗	✗	✗	✓	✗	✗	✓	✓
[11]	2019	✓	✓	✗	✗	✗	✗	✓	✗	✗	✓	✗
[126]	2019	✓	✓	✗	✗	✗	✗	✗	✗	✗	✗	✗
[127]	2019	✓	✓	✗	✗	✗	✗	✗	✓	✗	✗	✗
[125]	2019	✓	✓	✗	✓	✓	✗	✗	✗	✓	✗	✗
[128]	2019	✓	✓	✗	✗	✗	✗	✓	✗	✗	✓	✗
[129]	2020	✗	✓	✗	✓	✓	✗	✗	✓	✗	✓	✗
[130]	2020	✗	✓	✗	✗	✗	✗	✗	✗	✗	✓	✗
[131]	2020	✓	✓	✗	✓	✗	✗	✗	✗	✗	✗	✗
[132]	2020	✓	✓	✗	✗	✗	✗	✗	✗	✓	✗	✗
[133]	2021	✗	✓	✗	✗	✗	✗	✗	✓	✗	✗	✗
[134]	2021	✓	✓	✓	✗	✗	✗	✓	✓	✗	✓	✓

✓—parameter mentioned; ✗—parameter not mentioned.

**Table 17 sensors-22-08073-t017:** Key challenges and primary benefits of AI-based cardiology.

Ref #	Year	Findings	
		Key Challenges and Barriers for E-Cardiology with AI	Benefits of AI in E-Cardiology
[136]	2016	Safety and transparency Algorithmic fairness and biases, Complexity Data privacy and information security Need for infrastructure, High quality data Public perceptions about AI, Informed consent	Better diagnosis, better services Improves quality of services Time saving Reduced treatment cost
[137]	2019	Fitting confounders accidentally versus actual signal, Generalizability, Algorithmic bias Possibility of adversarial attack Logistical challenges in deploying AI systems Robust and rigorous quality assurance Traditional reluctance to switch from existing model to AI model in healthcare Algorithmic accountability To develop a relation between physicians and human-centered AI tools	N/A
[138]	2019	Data privacy, Accountability Algorithmic bias Adaptability, Complexity	Improved healthcare Better diagnosis High accuracy
[139]	2019	Privacy and discrimination Dynamic information and consent Transparency and ownership	Speedy imaging, Increased efficiency Greater insight into predictive screening Decreased healthcare cost
[140]	2020	Respect for autonomy Beneficence Non-maleficence and justice	Lower cost Improved diagnosis and treatment
[141]	2021	Safety, Privacy and security threats Ethical challenges Regulatory and policy challenges Availability of quality data and Lack of data standardization Distribution shifts Upgrading hospital infrastructure	Disease prediction and diagnosis Better image interpretation Real-time monitoring
[142]	2021	Sometimes data reflects inherent biases and disparities Huge dataset requirement Patient’s confidentiality Potential to be detrimental	Improved decision making Improved precision and predictability Intraoperative guidance via video

## Data Availability

Not applicable.

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
