# Peer review of "E-Cardiac Care: A Comprehensive Systematic Literature Review"

_sensors, 2022, doi:10.3390/s22208073_

Round 1

Reviewer 1 Report

The manuscript proposes to present a systematic review concerning E-cardiac care and Telecardiology within the context of IoT and Artificial Intelligence, which is a relevant field. Although the research field is significantly relevant, but the content is not new, and it does not bring novelty in relation to other Systematic Reviews. Concerning Medical IoT itself, we can find several Systematic Reviews with very similar goals and methodologies, such as: "A systematic review of IoT in healthcare: Applications, techniques, and trends", vol. 192, 2021, from Journal of Network and Computer Applications; and "IoT-Based Applications in Healthcare Devices, vol. 2021, 2021, from Journal of Healthcare Engineering (SI Medical Internet of Things (IoT) Devices). Therefore, it is not clear which the main differential of the proposed methodology. For example, is it just the focus at the Cardiological context ? 

Some information is not necessary for a Systematic Review and may be removed, such as the definition of cardiovascular diseases and types of arrhythmias which we found at Introduction.

The English writing should be carefully reviewed as we can find several verbal concordance erros and meaningless sentences. 

The authors say that the review study also provides recommendations and future guidelines/research directions for researchers and cardiologists, but the content related the referred future guidelines boils down to a few sentences (second paragraph from Discussion).

Reviewer 2 Report

Authors have presented their work on E-Cardiac Care and Telecardiology: A Comprehensive Systematic Literature Review. I feel that the manuscript is worthy to accept with the following revisions :

1. Authors shall improve the description of types of Arrhythmia.

2. Literature review section shall be improved.

3.What is the criteria to choose the Quality assessment parameters.

4.Figure 3 is found incomplete- check 

5. Oxygen sensory unit- few more works shall be reviewed .

6. Conclusion part is little weak comparing to the works done by authors- Check 

Round 2

Reviewer 1 Report

The authors achieved to address all the review requirements.
